# The Prediction of Quality of Life by Frailty and Disability among Dutch Community-Dwelling People Aged 75 Years or Older

**DOI:** 10.3390/healthcare12090874

**Published:** 2024-04-23

**Authors:** Robbert J. J. Gobbens, Tjeerd van der Ploeg

**Affiliations:** 1Faculty of Health, Sports and Social Work, Inholland University of Applied Sciences, 1081 HV Amsterdam, The Netherlands; tjeerd.vanderploeg@inholland.nl; 2Zonnehuisgroep Amstelland, 1186 AA Amstelveen, The Netherlands; 3Department Family Medicine and Population Health, Faculty of Medicine and Health Sciences, University of Antwerp, 2610 Antwerp, Belgium; 4Tranzo, Tilburg University, 5037 DB Tilburg, The Netherlands

**Keywords:** older people, quality of life, frailty, disability, activities of daily living, instrumental activities of daily living, WHOQOL-BREF

## Abstract

The present study aimed to examine the prediction of quality of life by frailty and disability in a baseline sample of 479 Dutch community-dwelling people aged 75 years or older using a follow-up period of 8 years. Regarding frailty, we distinguish between physical, psychological, and social frailty. Concerning physical disability, we distinguish between limitations in performing activities in daily living (ADL) and instrumental activities in daily living (IADL). The Tilburg Frailty Indicator (TFI) and the Groningen Activity Restriction Scale (GARS) were used to assess frailty domains and types of disability, respectively. Quality of life was determined by the WHOQOL-BREF containing physical, psychological, social, and environmental domains. In our study, 53.9% of participants were woman, and the mean age was 80.3 years (range 75–93). The study showed that psychological frailty predicted four domains of quality of life and physical frailty three. Social frailty was only found to be a significant predictor of social quality of life and environmental quality of life. ADL and IADL disability proved to be the worst predictors. It is recommended that primary healthcare professionals (e.g., general practitioners, district nurses) focus their interventions primarily on factors that can prevent or delay psychological and physical frailty, thereby ensuring that people’s quality of life does not deteriorate.

## 1. Introduction

The Department of Economic and Social Affairs of the United Nations forecasts that, worldwide, the number of people aged ≥65 years will rise from 761 million in 2021 to 1.6 billion in 2050. In addition, globally, the number of people aged ≥80 years is rising even faster (155 million in 2021 versus 459 million in 2050) [1]. This a result of many factors, e.g., longer life expectancy and low fertility [2]. Population aging creates multiple challenges in which a distinction can be made between the cultural challenges (ensuring that older people can live their lives with dignity and purpose), social challenges (optimising the age of retirement), and biological challenges (retaining a high level of mental and physical capacity) [3]. Moreover, population aging provides financial challenges, e.g., an increase in healthcare costs. In particular, developed countries are struggling to find money for retirement income [4].

To cope with the challenges presented by the aging population, policies and services in many countries in the world are focused on aging in place. The World Health Organization Centre for Health Development defines aging in place as “meeting the desire and ability of people through the provision of appropriate services and assistance to remain living relatively independently in the community in his or her current home or an appropriate level of housing” [5]. Because aging in place has become an important topic, it is essential to understand the quality of life of community-dwelling older people and the influencing factors of their quality of life.

Quality of life is increasingly an important concept in medical, psychological, and social studies [6]. It can be considered a highly relevant outcome measure when public policy is evaluated [7]. However, there is no consensus yet on defining it [8], though experts do agree that quality of life is a multidimensional and dynamic concept consisting of both objective and subjective components [6].

A definition of quality of life, which is often referred to, is developed by the World Health Organization. This definition emphasizes the subjective components of quality of life and states as follows: “An individual’s perception of their position in life and in relation to their goals, expectations, standards, and concerns” [9]. Based on a thematic synthesis of the perspectives of community-dwelling older people themselves, including 48 qualitative studies, nine quality-of-life domains were identified: role and activity, relationships, emotional comfort, financial security, autonomy, health perception, attitude and adaptation, home and neighborhood, and spirituality [10].

Several instruments are available for assessing quality of life, including specific measures designed to be relevant in certain subpopulations; for example, people with diabetes mellitus [11], or generic measures that can be used across a wide range of populations, including the Medical Outcomes Study 36-Item Short Form (SF-36) health survey [12] and the World Health Organisation Quality of Life—short form (WHOQOL-BREF) [13].

In the context of the aging population, frailty and disability are relevant concepts. After all, both are associated with advanced age and may lead to adverse outcomes among older people, such as an increase in healthcare utilization [14,15] and premature death [16,17,18,19].

Many experts indicate that disability is a potential consequence of frailty. The European, Canadian, and American Geriatric Advisory Panel argued that frailty should be considered as a predisability state [20]. This is confirmed by a more recent systematic review and meta-analysis, including 20 studies, showing that frailty in community-dwelling older people is a significant predictor of incident and worsening disability [21].

In addition to the aforementioned adverse outcomes (increase in healthcare utilization, premature death), frailty and physical disability are associated with lower quality of life among older people living in the community [22,23]. However, it should be noted that in the systematic review focusing on the associations between frailty and quality of life, 11 cross-sectional studies and only two longitudinal studies were included, and the corresponding meta-analysis consisted exclusively of cross-sectional studies (N = 4) [22]. Longitudinal studies focusing on the association between physical disability and quality of life among community-dwelling older people are even rarer. We found a 12-year longitudinal study aimed at identifying different patterns of development of quality of life among Chinese older people [24]. Those who did not exhibit any disability reported a higher level of quality of life over time.

The present study aimed to examine the prediction of quality of life by frailty and physical disability in a sample of Dutch community-dwelling people aged 75 years or older using a follow-up period of 8 years. Traditionally, frailty was defined primarily as having limitations in physical functioning. The phenotype of frailty by Fried et al. is a good example of this because it includes five physical criteria by which to determine whether an older individual is frail: low physical activity, weakness, unintentional weight loss, slow walking speed, and exhaustion [25]. We deliberately took a broad approach; so we were not only focused on physical frailty, but also on psychological and social frailty [26]. Focusing only on physical frailty potentially encourages a fragmentation of care. Also, previous studies have demonstrated that the effects of physical, psychological, and social frailty on quality of life are different. For example, in a cross-sectional study including 671 Dutch citizens aged ≥ 70 years, it was shown that feeling alone, a component belonging to social frailty, was the only component associated with all quality-of-life facets of the WHOQOL-OLD [27]. In another Dutch study, also using the WHOQOL-OLD, the correlations between the three frailty domains and the quality-of-life facets were not equally strong (e.g., physical frailty and sensory abilities: −0.462, *p*-value < 0.001 versus social frailty and sensory abilities: −0.189, *p*-value < 0.01) [28].

Concerning physical disability, we distinguish between limitations in performing activities in daily living (ADL) and instrumental activities in daily living (IADL). Examples of ADL are washing and drying your whole body, standing up and sitting in a chair, and getting in and out of bed. Doing “heavy” household activities (e.g., mopping, cleaning the windows, and vacuuming), making the beds, and doing the shopping are examples of IADL. The latter is considered a less severe form of disability [29]. This is supported by the finding in a Dutch study among 377 older people (aged ≥ 70 years) that 54.6% had at least one ADL disability and 67.4% had at least one IADL disability [23].

Our general hypothesis is that higher scores on the frailty and disability subscales lead to poorer quality of life as measured with the WHOQOL-BREF questionnaire.

## 2. Methods

### 2.1. Study Population and Data Collection

In June 2008, a sample of 1154 community-dwelling people aged 75 years or older was randomly drawn from the municipality of Roosendaal in the Netherlands, a municipality with 78,000 inhabitants. A questionnaire including the Tilburg Frailty Indicator (TFI) [26], the Groningen Activity Restriction Scale (GARS) [30,31], and questions about socio-demographic characteristics was sent to the people in the sample. A total of 484 people completed the questionnaire. For the TFI and the GARS, see Appendix A and Appendix B, respectively. For 8 consecutive years, the people who belonged to the sample were asked to complete the same questionnaire. In this study, we will present the results of five measurements (baseline, 2 years later, 4 years later, 6 years later, and 8 years later). The sample was previously mainly used for frailty studies, e.g., focusing on the psychometric properties of the TFI [26,32].

### 2.2. Frailty Measurements

The TFI was used to assess frailty of older people [26]. In this study, we used the data from five determinants in Part A of the TFI: gender, age, education, income, and multimorbidity. Furthermore, we used the data of 15 components of frailty of Part B of the TFI. These 15 components are distributed over four domains: physical frailty (eight components), psychological frailty (four components), and social frailty (three components). The components of physical frailty are being physically unhealthy, unexplained weight loss, difficulty in walking, difficulty in maintaining balance, poor hearing, poor vision, lack of strength in the hands, and physical tiredness. The components of psychological frailty are having problems with memory, feeling down, feeling nervous or anxious, and being unable to cope with problems. Finally, social frailty includes living alone, lack of social relations (loneliness), and lack of social support. The scores range from 0 to 8 for the physical, 0 to 4 for the psychological, and 0 to 3 for the social domains of frailty. Higher scores reflect a higher level of frailty. The TFI has demonstrated good psychometric properties among Dutch community-dwelling older people [26,32]. For the questionnaire, see Appendix A.

### 2.3. Disability Measurements

The GARS, a self-reported questionnaire, was used to assess disability among older people. The GARS contains two subscales: the activities of daily living (ADL) subscale, with 11 items, and the instrumental activities of daily living (IADL) subscale, with seven items. Each of the 18 items has four ranked response categories, indicating the extent to which the respondent has difficulty performing an activity. The scores on the GARS ADL subscale and IADL subscale range from 11 to 44 and 7 to 28, respectively. Higher scores refer to more ADL and IADL disability. The GARS has been validated in the Netherlands and demonstrated to have good psychometric properties to assess disability among older people [30,31]. For the questionnaire, see Appendix B.

### 2.4. Quality of Life Measurements

The WHOQOL-BREF questionnaire was used to assess the health-related quality of life of older people [13]. This questionnaire was developed as an abbreviated version of the WHOQOL-100 for use in situations in which the burden of the respondent burden should be as limited as possible, and time is restricted [33]. This WHOQOL-BREF consists of 26 questions. The first two questions refer to the overall quality of life and satisfaction with general health. The remaining 24 questions, which we have used in the present study, are distributed over four domains: physical health (seven questions), psychological (six questions), social relations (three questions), and environmental (eight questions). Each question features a Likert scale ranging from one to five. The total score for each domain is calculated as the mean of the responses to the underlying questions multiplied by 4. Higher scores indicate a higher quality of life in each domain. Many studies have demonstrated that the WHOQOL-BREF has good properties in determining the quality of life among people aged ≥ 50 years [33,34,35]. For the questionnaire, see Appendix C.

### 2.5. Outcomes, Predictors, and Adjustment Variables

The repeated quality-of-life-domain scores (physical, psychological, social, environmental) at the five time points (at baseline, 2 years later, 4 years later, 6 years later, and 8 years later) were used as outcome variables. We used the frailty domain scores (physical frailty, psychological frailty, and social frailty) and the scores on types of disability (ADL and IADL) at baseline as predictor variables. The variables gender, age, education, income, and multimorbidity from Part A of the TFI (see Appendix A) acted as adjustment variables.

### 2.6. Ethical Considerations

For this study, medical ethics approval was not necessary as particular treatments or interventions were not offered or withheld from respondents. The integrity of the respondents was not encroached upon as a consequence of participating in the present study, which is the main criterion in medical–ethical procedures in the country where this study was conducted (The Netherlands) [36]. The study was conducted according to the guidelines for good clinical practice. The researchers did not make the questionnaire long, so the burden on participants (people aged 75 years or older) was limited; the average time for completing the questionnaire was only about 20 min. In addition, the questionnaire contained measures (WHOQOL-BREF, TFI, GARS) that have already been used in many previous studies among older people [32,37,38,39,40], and from this, it was found that the target group is perfectly capable of completing these measures.

### 2.7. Statistical Analysis

The internal consistency of the items within the quality-of-life-domain scores (physical, psychological, social, environmental) at each of the five time points was measured with the Cronbach alpha statistic. A value of Cronbach alpha *>* 0.60 was considered as an indication of consistency [41]. Generalized estimating equations (GEE) is a statistical technique used to analyze correlated response data [42,43]. GEE is particularly useful in medical, social sciences, and behavioral sciences research, where subjects are followed over time, and measurements are taken at several time points. In our study, GEE was used for the analysis of the outcome variables (quality-of-life-domain scores) over time, first with just the predictor variables (frailty domains, types of disability) and then with the predictor variables and the adjustment variables (gender, age, education, income, multimorbidity) together. For the GEE analysis, we used the settings “exchangeable” for the correlation structure and “Gaussian” for the distribution family. For all analyses, we used R version 4.1.2 [44]. A *p*-value < 0.05 was considered significant.

## 3. Results

This section starts with the results of the consistency analyses of the items within the quality-of-life-domain scores (physical, psychological, social, environmental) at each of the five time points (Table 1). The distribution (in percentages) of the adjustment variables, gender, age, education, income, and multimorbidity, at the five time points is presented in Table 2. The results of the GEE analyses (unadjusted and adjusted) for the four quality-of-life outcomes (physical, psychological, social, environmental) with the predictors and the adjustment variables are presented in Table 3, Table 4, Table 5 and Table 6.

For the four quality-of-life-domain scores, at each time point, the value of the Cronbach alpha was >0.6 (Table 1). This result justifies the use of mean domain scores based on the item scores within each domain.

Table 2 shows the distribution (in percentages) of the adjustment variables at each time point as they were used in the GEE analyses. At all five time points, more women than men participated in our study. At T = 0, the mean age of the participants was 80.3 years (range 75–93). At T = 8, the percentage of people aged 80 years or older is lower than at T = 0 (32.1% versus 47.3%). This also applies to the percentage of older people with multimorbidity (32.1% at T = 8 versus 46.4% at T = 0). However, the differences overall in percentages are not substantial.

Table 3 shows the results of the GEE analysis (unadjusted and adjusted) for the “Physical QoL score”. All predictor variables, except “social frailty score”, showed *p*-values < 0.05 after adjustment.

The results of the GEE analysis for the “Psychological QoL score” are presented in Table 4. Both the physical and psychological frailty scores were significant (unadjusted and adjusted) (*p*-values < 0.001). Also, the adjusted “ADL disability score” was significant (*p*-value 0.006). The predictor variables “social frailty score” and “IADL disability score” showed *p*-values ≥ 0.05 (unadjusted and adjusted) and were, therefore, not significant.

Regarding the outcome “social QoL score”, the variables “psychological frailty score” and “social frailty score” showed *p*-values < 0.05 after adjustment, see Table 5. None of the disability types were found to be a significant predictor of the “social QoL score”.

For the outcome “environmental QoL score”, all frailty scores are significant after adjustment (*p*-values < 0.05), see Table 6. The ADL and IADL scores were not significant predictors of the “environmental QoL score”.

## 4. Discussion

With the aging population and the policy focused on aging-in-place quality of life among community-dwelling older people is an important issue these days. Because frailty and disability are also more common in an aging population, we aimed to examine the prediction of quality of life among community-dwelling older people by three frailty domains (physical, psychological, social) and two types of disability (ADL, IADL), assessed by the validated self-report questionnaires TFI [26] and GARS [30,31]. Gender, age, education, income, and multimorbidity acted as adjustment variables. We used a Dutch sample consisting of 479 individuals aged 75 years or older. Following a baseline measurement, we conducted measurements on them after 2, 4, 6, and 8 years. We hypothesized that higher scores on the frailty and disability subscales lead to poorer quality of life.

Our study showed that psychological frailty predicted all four domains of quality of life (physical health, psychological, social relations, environmental) assessed with the self-report questionnaire WHOQOL-BREF significantly, unadjusted, and adjusted. Seven of the eight *p*-values were <0.001. Physical frailty predicted three quality-of-life domains significantly, unadjusted, and adjusted: physical quality of life, psychological quality of life, and environmental quality of life. Social frailty was the worst predictor. This frailty domain was only found to be a significant predictor of social quality of life and environmental quality of life, unadjusted and adjusted. Regarding the two types of disability, ADL disability was found to be a significant predictor of physical quality of life (unadjusted and adjusted) and psychological quality of life (adjusted). IADL disability predicted physical quality of life significantly (unadjusted and adjusted).

Our main finding is that the psychological frailty domain predicted all four quality-of-life domains of the WHOQOL-BREF. This finding cannot be confirmed by other studies. Only one validation study, using the same sample, is known to have shown that psychological frailty did not predict the domains of the WHOQOL-BREF using regression analysis and a follow-up period of 2 years [32]. However, in a cross-sectional study including a sample of 257 Greek older people (aged > 60 years), psychological frailty assessed with the TFI was significantly correlated with all domains of the WHOOQL-BREF (all *p*-values < 0.001) [45]. Psychological frailty operationalised according to the TFI consists of four components: problems with memory, feeling down, feeling nervous or anxious, and feeling unable to cope with problems. Studies have been conducted on the predictive value of these individual frailty components for the quality of life of older people. A Dutch cross-sectional study among 1031 people aged ≥ 65 years showed that both frailty domains assessed with the TFI contain several components associated with the four quality of life domains of the WHOQOL-BREF significantly [46]. The psychological frailty component “feeling down” was associated with all four domains, even after adjusting for the effects of all other variables in the model, including all other frailty components of the TFI and background characteristics of the participants. Feeling down refers to depression, which was also found to be associated with poor quality of life independently of physical frailty in a cross-sectional study on community-dwelling older people with reference to an outpatient geriatric service in Italy [47]. The TFI psychological frailty component “feeling nervous or anxious” is also known to be associated with scores on quality-of-life domains [27,29]. In the study referenced earlier, this component had significant effects on the physical, psychological, and environmental quality-of-life domains of the WHOQOL-BREF [46]. We recommend future longitudinal studies aimed at examining the prediction of quality of life by psychological frailty using the WHOQOL-BREF.

In contrast to the lack of studies aimed at predicting quality of life by psychological frailty, the prediction of quality of life by physical frailty has often been a subject of study. Our finding that physical frailty predicted quality of life is supported by a meta-analysis including four cross-sectional studies showing that physical frailty operationalised according to the phenotype of frailty by Fried et al. [25] was associated with lower mental and physical quality-of-life scores on the SF-36 [22]. Of the physical frailty components, the components “physical inactivity” and “physical tiredness”, in particular, are very decisive for older people’s quality of life assessed with the WHOQOL-BREF [46] but also assessed with the WHOQOL-OLD [27].

As mentioned in the introduction, longitudinal studies focusing on the prediction of quality of life by physical disability are scarce. As a result, we can only compare our findings with results from cross-sectional studies. In Germany, ADL and IADL disability were only associated with the domain “physical health” of the WHOQOL-BREF in subjects aged ≥70 years [48]. In a Dutch sample of 377 individuals aged 75 years or older, ADL disability and IADL disability significantly explained the variance of the score on the physical and mental dimensions of quality of life [23], assessed with the Short-Form Health Survey (SF-12) [49]. In China, older people with limitations in mobility reported a lower quality of life [50], assessed with the Five-Dimensional European Quality of Health Scale (EQ-5D).

Our study has shown that psychological frailty and physical frailty are important predictors of life among community-dwelling older people. The components of these domains of frailty constitute focal points to intervene aimed at preventing or delaying lower quality of life. In a systematic review and meta-analysis of controlled trials focused on the effectiveness of psychosocial services for depression and anxiety (two components of psychological frailty) in Chinese older people, it was observed that an overall significant treatment effect was present; in-person and home-based interventions provided by nurse practitioners appeared to be statistically significant [51]. A more recent systematic review and meta-analysis aiming to evaluate the efficacy of telemedicine interventions to reduce depression and anxiety demonstrated that telemedicine interventions are feasible, and improvement in depression or anxiety was demonstrated in multiple studies [52]. Many studies have been conducted on interventions aimed at preventing or delaying the occurrence of physical frailty components. Based on 10 cohort studies, it was concluded that a higher level of physical activity was associated with lower odds of physical frailty and multidimensional frailty, including psychological and social components of frailty [53]. It also appears, although still with limited evidence, that volunteer-led interventions, including resistance exercise training, can improve frailty status among community-dwelling older people [54]. Another important component for the development of physical frailty is nutrition; this involves both quantitative (energy intake) and qualitative (nutrient quality) factors [55]. For example, multi-nutrient supplementations significantly improved handgrip strength [56]. Healthcare professionals, such as general practitioners, nurses, and physical therapists, should be knowledgeable about the effectiveness of interventions and should apply proven effective interventions in their practice to prevent or delay frailty so that the quality of life of older people does not deteriorate further. Since frailty can be considered a precursor of disability [20], interventions targeting frailty may also contribute to preventing and delaying disability, thereby potentially preventing or delaying well-known adverse outcomes of disability such as increased healthcare utilization and premature death [14,17].

Some limitations of our study should be noted. Firstly, the TFI and the GARS were used to assess physical, psychological, and social frailty and ADL and IADL disability, respectively. Both are self-report scales that lack performance-based measures. Integrating both self-report and performance-based measures could offer a more comprehensive understanding of both concepts. However, a cross-sectional study including a sample of 135 people (mean age 73.8 years; SD 7.0) suggested substituting performance-based criteria with self-report questions in defining the frailty phenotype by Fried et al. [25] due to the substantial agreement between the two measures [57]. In another study among 349 individuals aged 80 years or older, it was found that self-reported ADL and IADL disability closely corresponded with performance-based measurements [58]. Secondly, at baseline, the sample comprised 479 older individuals, aged ≥ 75 years, with a mean age of 80.3 years (SD 3.8). However, a considerable number of participants were lost during the 8-year follow-up period. This has implications for the generalizability of the findings to the wider Dutch population and their external validity. In a prior study, it was demonstrated that 162 individuals from this cohort passed away between the years 2008 and 2015. Notably, both frailty and disability, evaluated using the TFI and GARS, respectively, were predictive of mortality [16,17]. Thirdly, regarding the multivariable analysis, adjustments were made for gender, age, education, income, and multimorbidity. The inclusion of other variables in this analysis (e.g., ethnicity, marital status) could potentially yield different outcomes.

In addition to these limitations, our study also boasts several strengths. Notably, we employed three widely recognized instruments (TFI, GARS, and WHOQOL-BREF) to assess frailty, disability, and quality of life, respectively. The psychometric properties, including reliability and validity, of these instruments are robust [26,30,31,32,33,34,35]. Moreover, we conducted five measurements among individuals aged 75 years or older at baseline, enabling us to perform a Generalized Estimating Equations (GEE) analysis spanning 8 years.

## 5. Conclusions

In this study among Dutch community-dwelling people aged 75 years or older, we showed that psychological and physical frailty in particular, assessed with the TFI, predicted quality of life, measured with the WHOQOL-BREF. The prediction of quality of life by social frailty and the two types of disability (ADL, and IADL) proved to be much more limited. Therefore, we recommend that healthcare professionals focus their interventions primarily on factors that can prevent or delay psychological and physical frailty, thereby ensuring that people’s quality of life does not deteriorate.

## Figures and Tables

**Table 1 healthcare-12-00874-t001:** Cronbach alpha.

Domain	T = 0	T = 2	T = 4	T = 6	T = 8
Physical	0.87	0.86	0.88	0.85	0.83
Psychological	0.77	0.79	0.78	0.72	0.80
Social	0.63	0.67	0.71	0.68	0.73
Environmental	0.79	0.83	0.84	0.82	0.74

**Table 2 healthcare-12-00874-t002:** Adjustment variables per time point (%).

	T = 0 (n = 347)	T = 2 (n = 197)	T = 4 (n = 155)	T = 6 (n = 103)	T = 8 (n = 81)
Gender					
Man	46.1	47.7	45.2	48.5	46.9
Woman	53.9	52.3	54.8	51.5	53.1
Age					
Younger than 80	52.7	54.3	58.7	65.0	67.9
80 or older	47.3	45.7	41.3	35.0	32.1
Education					
Primary or secondary	83.6	80.7	80.0	77.7	77.8
Higher	16.4	19.3	20.0	22.3	22.2
Income in euro					
1800 or less	69.5	69.0	65.2	62.1	61.7
More than 1800	30.5	31.0	34.8	37.9	38.3
Multimorbidity					
None or one	53.6	56.9	60.6	57.3	67.9
Two or more	46.4	43.1	39.4	42.7	32.1

**Table 3 healthcare-12-00874-t003:** Physical QoL score.

	Unadjusted		Adjusted	
	Coefficient	*p*-Value	Coefficient	*p*-Value
Intercept	19.63	<0.001	19.83	<0.001
Frailty score				
Physical health	−0.63	<0.001	−0.57	<0.001
Psychological	−0.49	<0.001	−0.41	<0.001
Social relations	−0.13	0.174	−0.14	0.155
Disability score				
ADL	−0.14	<0.001	−0.17	<0.001
IADL	−0.07	0.003	−0.06	0.033
Adjustment variables				
Gender			0.19	0.314
Age			0.14	0.413
Education			0.10	0.690
Income			0.05	0.815
Multimorbidity			−0.62	0.002

**Table 4 healthcare-12-00874-t004:** Psychological QoL score.

	Unadjusted		Adjusted	
	Coefficient	*p*-Value	Coefficient	*p*-Value
Intercept	17.52	<0.001	17.37	<0.001
Frailty score				
Physical health	−0.23	<0.001	−0.23	<0.001
Psychological	−0.77	<0.001	−0.72	<0.001
Social relations	−0.03	0.705	−0.09	0.327
Disability score				
ADL	−0.04	0.067	−0.07	0.006
IADL	−0.02	0.282	−0.01	0.563
Adjustment variables				
Gender			0.45	0.003
Age			0.18	0.244
Education			0.36	0.068
Income			0.05	0.789
Multimorbidity			0.10	0.529

**Table 5 healthcare-12-00874-t005:** Social QoL score.

	Unadjusted		Adjusted	
	Coefficient	*p*-Value	Coefficient	*p*-Value
Intercept	16.33	<0.001	16.32	<0.001
Frailty score				
Physical health	−0.22	0.006	−0.16	0.071
Psychological	−0.47	<0.001	−0.45	0.002
Social relations	−0.72	<0.001	−0.98	<0.001
Disability score				
ADL	0.06	0.175	−0.02	0.627
IADL	0.01	0.836	0.05	0.234
Adjustment variables				
Gender			1.62	<0.001
Age			0.47	0.070
Education			0.23	0.503
Income			−0.19	0.523
Multimorbidity			−0.42	0.134

**Table 6 healthcare-12-00874-t006:** Environmental QoL score.

	Unadjusted		Adjusted	
	Coefficient	*p*-Value	Coefficient	*p*-Value
Intercept	17.91	<0.001	17.24	<0.001
Frailty score				
Physical health	−0.29	<0.001	−0.28	<0.001
Psychological	−0.51	<0.001	−0.45	<0.001
Social relations	−0.34	0.001	−0.38	<0.001
Disability score				
ADL	−0.03	0.313	−0.05	0.131
IADL	−0.01	0.676	0.01	0.837
Adjustment variables				
Gender			0.60	0.002
Age			0.09	0.592
Education			0.46	0.039
Income			0.74	<0.001
Multimorbidity			0.02	0.900

## Data Availability

The data presented in this study are available on request from the corresponding author.

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
