# Peer review of "The Prediction of Quality of Life by Frailty and Disability among Dutch Community-Dwelling People Aged 75 Years or Older"

_healthcare, 2024, doi:10.3390/healthcare12090874_

Round 1
Reviewer 1 Report
Comments and Suggestions for Authors
Dear authors, thanks for the opportunity to review this primary research. Although you have done excellent work, some issues should be addressed.
1. SCOPE: the manuscript is in line with the thematic scope of the Healthcare.
2. TITLE: is suitable!
3. ABSTRACT: the length and quality are correct.
- [lines 13-25]: to be more interesting in this abstract, I would suggest presenting the findings more comprehensively and please avoid copy-paste specific parts of “text” from the main body. For instance, some results and the entire conclusion.
4. KEYWORDS: The number of keywords should be at least six (6) – please add some more (eg. ADL, IADL…).
5. INTRODUCTION: the introduction section is not clear and well-structured!
- [lines 53-66]: please focus on the description of the “problem” (what is already known prediction of QoL by frailty? Why is so important to examine OoL domains (physical, psychological, social) as predictors?
- [lines 53-66]: Also, authors should avoid mentioning the properties of specific screening tools for the assessment of the QoL. I would suggest removing these two paragraphs and presenting some evidence showing the importance of the usage of “frailty and disability predictors” in the QoL measurement particularly in the ELDERLY, due to the absence of consensus definitions of frailty (clinical criteria) and several other factors such as multimorbidity and individual characteristics!
- [lines 87: Authors should clearly state the “problem” before the main question (aim). Authors should present the research gap in knowledge and the importance of the new evidence using frailty and disability to predict “poor’ or “good” QoL for seniors!
- [lines 89-111]: Similarly, authors should describe the problem (gap) summarizing the reason why this study is so important to be conducted and not provide the “study analysis”. For this, I would suggest moving this valuable information [lines 89-111] in a subsection in METHODS as 2.2 “Study analysis” explaining the rationale for follow-up and the selected variables of interest … and the subsection “Frailty Measurements as ” 2.3.
6. METHOD: the structure should be revised
- [line 125]: see my comment [I would suggest moving this valuable information [lines 89-111] in a subsection in METHODS as 2.2 “Study analysis” explaining the rationale for follow-up and the selected variables of interest … and the subsection “Frailty Measurements as ” 2.3.]
- [lines 173]: usually, the “statistical analysis” subsection is presented at the bottom of METHODS preluding the RESULTS section! (structural comment)!
7. RESULTS: the results of the focus study are properly presented and well-described.
- The presented data and findings are sufficient to conclude.
- Line 206: the proportions are seems to be reversed for the [Table 1: multi-morbidity for T0 & T8 ]
- Lines 212-241: In Tables 2-5: the Adjustment's Coefficients seems to be to another column - it seems that the Tables 2-5 needs modification!
8. DISCUSSION: the results are discussed comprehensively even though only a few studies are available for comparison.
- I would suggest to authors to examine the following papers in case some information should be discussed!
- https://www.ncbi.nlm.nih.gov/pmc/articles/PMC9819361/
- https://pubmed.ncbi.nlm.nih.gov/33680593/
9. CONCLUSIONS: This study provides an important and sufficient conclusion!
- A minor suggestion: In my point of view, since the results were obtained from five measurements for eight years, this study should also highlight “physical frailty” as the main predictor taking into consideration that social roles may be lost during these 8 years.
- Please, avoid mentioning the screening tools and variables assessed in this section (e.g. WHOQOL-BREF, ADL…). [only a pure text]
10. REFERENCES: can be improved
Reviewer 2 Report
Comments and Suggestions for Authors
The present study aimed to examine the prediction of quality of life (physical, mental and social) in a sample of people aged 75 years or older. The topic is interesting because we live with an aging population and if we investigate which variables influence their quality of life, we can create programs that improve their lives. Despite this, I believe that there are some nuances to improve within this paper.
1. Add % women and men from the sample in the abstract and participants section. As well as the average and age range.
2. There is talk of 5 moments of taking measurements and it is observed that the n is decreasing. However, the reasons for this decrease are not specified. Specify main reasons and % of women and men who are "falling."
3. Specify in the measurements section which sociodemographic questions were asked.
4. Calculate the Crombash alpha for this dead body for each measurement used.
5. The general objective is very clear but the different hypotheses from which the research team is based are not specified, these must be specified.
6. In the results and discussion section, follow the same order of presentation of objectives and hypotheses to better understand everything.
7. In most of the results it is observed that there are differences in gender, it would be good to see what these differences may be due to, if there are different predictors for men and women.
